

# High incidence of plasmid-mediated quinolone resistance genes among ciprofloxacin-resistant clinical isolates of Enterobacteriaceae at a tertiary care hospital in Puducherry, India

Thiyagarajan Yugendran[1] and Belgode Narasimha Harish[1]

Department of Microbiology, Jawaharlal Institute for Post-graduate Medical Education and Research, Puducherry, India

## ABSTRACT

**Background.** Plasmid-mediated quinolone resistance (PMQR) has received considerable attention recently. Data analysis in Jawaharlal Institute of Postgraduate Medical Education & Research (JIPMER) revealed 75% of the Enterobacteriaceae isolates to be ciprofloxacin-resistant in 2012. Few reports regarding the prevalence of PMQR are available from India. Hence, the present study was carried out to ascertain the prevalence of PMQR genes among clinical isolates of ciprofloxacin-resistant Enterobacteriaceae in JIPMER.

**Methods.** The study included 642 ciprofloxacin-resistant clinical Enterobacteriaceae isolates. JIPMER hospital's annual consumption data for fluoroquinolones were retrieved from the Department of Pharmacy. The test isolates were screened for the presence of *qnr A*, *B*, *D*, *S* and *aac*(6′)-*Ib-cr* genes. PMQR-positive isolates alone were tested for the presence of class I (*intI1*) and class II (*intI2*) integrons. Randomly selected PCR amplicons were sequenced and analysed using MEGA software. A total of 30 PMQR strains chosen at random were assessed for the transferability of the PMQR genes.

**Results.** A majority of the strains exhibited high MIC values with 106 strains exhibiting MIC values >256 μg/mL. The *aac*(6′)-*Ib-cr* gene had the highest prevalence at 64% (414) while, *qnrB* and *qnrS* genes were present in 15% (97) and 10% (64) of the isolates respectively. None of the strains were positive for *qnrA* and *qnrD*. All PMQR-positive isolates were screened for class I (*intI1*) and class II (*intI2*) integrons. Class I integron was found to be predominant among the test isolates with a few of them carrying both the classes of integrons. Transferability of PMQR genes to transconjugants was identified.

**Conclusion.** The incidence of PMQR genes in the tertiary-care setup of the JIPMER hospital was found to be high which could be probably due to the increased prescription of fluoroquinolones. Thus, there is a need for rational usage of fluoroquinolones.

Corresponding author
Belgode Narasimha Harish,
drbnharish@gmail.com

## INTRODUCTION

Fluoroquinolone resistance among the Enterobacteriaceae is a serious public health problem as it is responsible for the rise in respiratory tract infections worldwide whereas, in Asia UTI and intra-abdominal infections are on the rise (*Dalhoff, 2012*). Accumulation of mutations within target DNA gyrase enzyme was thought be the only mechanism of fluoroquinolone resistance until in the year 1998 when *qnr*, a plasmid-borne fluoroquinolone resistance gene, was reported (*Martínez-Martínez, Pascual & Jacoby, 1998*). Similarly, there are a few more reports on fluoroquinolone resistance mechanism/s that involve enzymatic modification of fluoroquinolones by *aac*(6′)-*Ib*-*cr* encoded Fluoroquinolone-modifying enzyme and efflux pump activities mediated by *qepA* & *oqxAB* (*Robicsek, Jacoby & Hooper, 2006*; *Vetting et al., 2008*; *Robicsek et al. (2006)*; *Strahilevitz et al., 2009*). Of late, Enterobacteriaceae isolates have exhibited a high level of fluoroquinolone resistance (*Redgrave et al., 2014*). Due to the increase in fluoroquinolone resistance, plasmid-mediated quinolone resistance (PMQR) has received considerable attention in recent years. The *qnr* gene alleles *A*, *B*, *C*, *D* and *S*, encode for a pentapeptide repeat protein capable of protecting DNA gyrase from fluoroquinolones (*Robicsek et al., 2006*; *Strahilevitz et al., 2009*). Integrons are mobile genetic elements that have been identified in plasmids harbouring PMQR genes allowing them to spread horizontally for which they are widely feared (*Pazhani et al., 2011*).

PMQR genes have been stressed upon in many studies (*Strahilevitz et al., 2009*; *Pazhani et al., 2011*; *Mendez et al., 2009*). The quinolone resistance rate in clinical isolates of gram-negative bacilli ranges from 65% to 70% in India (*Hariharan et al., 2015*; *Varughese & Beniwal, 2015*). Despite such high fluoroquinolone resistance rates, fewer than ten studies from India have reported the prevalence of PMQR genes. (*Pazhani et al., 2011*; *Geetha et al., 2014*; *Magesh et al., 2011*; *Bhattacharya et al., 2011*). Data analysis in the diagnostic microbiology laboratory attached to Jawaharlal Institute of Postgraduate Medical Education & Research (JIPMER), Puducherry, India, a teaching hospital with a bed strength of 1,500, revealed 75% of the Enterobacteriaceae isolates to be resistant to ciprofloxacin in the year 2012 (T Yugendran, 2013, unpublished data). Therefore, in the present study ciprofloxacin-resistant isolates belonging to the family Enterobacteriaceae from the samples of patients attending JIPMER hospital were collected and screened for PMQR determinants and integrons with an aim to ascertain the PMQR prevalence in the hospital.

## MATERIALS & METHODS

The study was approved by JIPMER Institute Ethics Committee, Jawaharlal Institute of Post Graduate Medical Education and Research, Puducherry, India (ECR/324/Inst/PY/2013).

### Bacterial strains

A total of 642 clinical isolates belonging to the family Enterobacteriaceae resistant to ciprofloxacin by Kirby-Bauer disc diffusion method subsequently confirmed by agar dilution MIC were part of the study. Only one positive culture per patient was included. Standard methods were followed for isolation and identification of the bacteria from clinical specimens like blood, pus, CSF, etc. (*Forbes, Sahm & Weissfeld, 2007*). *E. coli* (J53),

*Shigella boydii* (IDH738), *E. coli* (BCH1108), *Morganella morganii* (500914) and *E. coli* (TC145) harbouring *qnrA*, *qnrB*, *qnrS*, *qnrD* & *aac(6')-Ib* and *qnrA* & *aac(6')-Ib-cr* were used as positive controls in the PCR assay. The ATCC strain *E. coli* 25922 served as the quality control in the antimicrobial susceptibility test.

## Antibiotic susceptibility test

The discs for amikacin (30 µg), ceftriaxone (30 µg), ceftazidime (30 µg), ciprofloxacin (5 µg) and gentamicin (10 µg), meropenem (10 µg) were prepared in-house using antibiotics in pure power form (Himedia Laboratories, Mumbai, India) whereas cefoperazone-sulbactam disc was procured from Microxpress Tulip Diagnostics Pvt. Ltd, India. In particular cases, bacteria were also tested for imipenem (Himedia Laboratories, Mumbai, India) and piperacillin-tazobactam (Microxpress Tulip Diagnostics Pvt. Ltd, Goa, India). The antibiotic susceptibility of the test isolates was interpreted as per CLSI M100-S25. Similarly, Minimum Inhibitory Concentration (MIC) values were determined according to CLSI M100-S25 guidelines for ciprofloxacin alone by agar dilution method using Muller Hinton agar procured from Himedia Laboratories, India. *E. coli* ATCC 25922 was included as the quality control. The lowest concentration of antibiotic at which the growth of bacteria had been completely inhibited was recorded as the MIC.

## Fluoroquinolone prescribing data

JIPMER hospital's annual prescribing data for fluoroquinolones were retrieved from Department of Pharmacy, JIPMER.

## PCR assay

DNA templates were prepared from the overnight inoculum of test strains grown on Nutrient HiVeg™ Agar (Himedia Laboratories, Mumbai, India) resuspended in MiliQ water after three rounds of washing. Crude template DNA was prepared by boiling lysis method (*Pazhani et al., 2011*). The reactions were performed in Flexilid Mastercycler PCR system (Eppendorf, Germany). The target genes, primer sequences, PCR conditions and amplified product sizes are given in Table 1. PMQR-positive isolates alone were screened for the class I (*intI1*) and class II (*intI2*) integrons. Electrophoresis and staining analysed the PCR products with ethidium bromide.

## Nucleotide sequencing

Sequencing of PCR products was carried out at Xcelris Genomics, Ahmedabad. Nucleotide sequences were analysed over BLAST server (http://blast.ncbi.nlm.nih.gov/Blast.cgi) against the GenBank database of the National Center for Biotechnology Information. The phylogeny of the randomly sequenced *aac(6')-Ib-cr* amplicons were determined based on the nucleotide substitutions per site, i.e., the number of 'nucleotide substitutions' within the given sequence divided by the length of the sequence using MEGA software (*Kumar, Stecher & Tamura, 2016*).

## Conjugation assay

PMQR-positive strains numbering 30 (*E.coli*-15, *Klebsiella pneumoniae*-4, *Klebsiella oxytoca*-3, *Enterobacter spp.*-3, *Citrobacter freundii*-3, *Proteus mirabilis*-2) were randomly

**Table 1  List of the primer-pairs used in this study.**

| Gene | | Primer sequence (5′-3′) | Amplicon size (bp) | Reference | PCR condition | |
|------|--|------------------------|-------------------|-----------|--------------|--|
| | | | | | $T^a$ | $T^b$ |
| qnrA | Forward | CAGCAAGAGGATTTCTCACG | 630 | *Ciesielczuk et al. (2013)* | 58 | 30 |
| | Reverse | AATCCGGCAGCACTATTACTC | | | | |
| qnrB | Forward | GGCTGTCAGTTCTATGATCG | 488 | *Ciesielczuk et al. (2013)* | 59.1 | 30 |
| | Reverse | SAKCAACGATGCCTGGTAG | | | | |
| qnrD | Forward | CGAGATCAATTTACGGGGAATA | 581 | *Cavaco et al. (2009)* | 57 | 30 |
| | Reverse | AACAAGCTGAAGCGCCTG | | | | |
| qnrS | Forward | GCAAGTTCATTGAACAGGGT | 428 | *Cattoir et al. (2007)* | 55.6 | 30 |
| | Reverse | TCTAAACCGTCGAGTTCGGCG | | | | |
| aac(6′)-Ib-cr | Forward | TTGGAAGCGGGGACGGAM | 260 | *Wareham et al. (2010)* | 58 | 30 |
| | Reverse | ACACGGCTGGACCATA | | | | |
| intI1 | Forward | GTTCGGTCAAGGTTCTG | 920 | *Pazhani et al. (2011)* | 55 | 45 |
| | Reverse | GCCAACTTTCAGCACATG | | | | |
| intI2 | Forward | ATGTCTAACAGTCCATTTT | 420 | *Pazhani et al. (2011)* | 55 | 30 |
| | Reverse | AAATCTTTAACCCGCAAAC | | | | |

**Notes.**
[a] Annealing temperature in °C.
[b] Extension time in s.

selected for assessing the transferability of the PMQR genes following a previously described method (*Jacoby & Han, 1996*) with *E. coli* (J53) AziR (sodium azide-resistant) as the recipient strain. Transconjugants were selected on MacConkey agar containing sodium azide (100 μg/mL) and ciprofloxacin (0.5 μg/mL) and confirmed based on the results of biochemical and antimicrobial susceptibility tests carried out for transconjugants, recipient and donor bacterial cells. Screening of the transconjugants by PCR assay determined the transferability of PMQR genes.

## RESULTS

### Antibiotic susceptibility test

Out of 642 Enterobacteriaceae isolates, 43 isolates were resistant to all the antibiotics tested. Resistance to ceftriaxone, ceftazidime and cefoperazone-sulbactam were seen in 398, 381 and 351 isolates respectively. All the isolates were resistant to ciprofloxacin. Resistance to ciprofloxacin was confirmed by the agar dilution method. The MIC values against ciprofloxacin ranged from 2 μg/mL to >256 μg/mL as summarised in Fig. 1. It is notable that 106 (~16%) strains had MIC values >256 μg/mL. A total of 112 isolates were resistant to meropenem, and it was most effective among all the tested antibiotics.

### Fluoroquinolone prescribing data

Data from the Department of Pharmacy revealed that ciprofloxacin is the most extensively used fluoroquinolone for various infections in JIPMER hospital followed by levofloxacin and ofloxacin (Data S1).

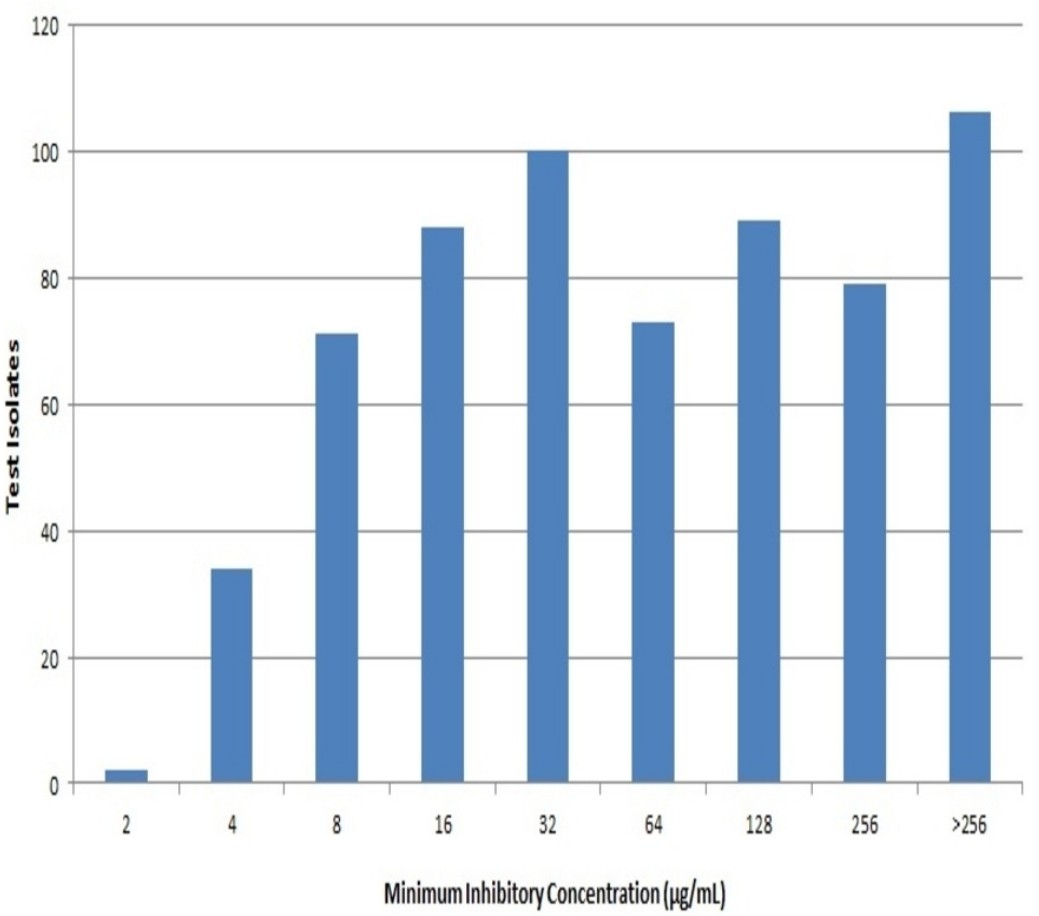

**Figure 1** Distribution of Minimum Inhibitory Concentration (MIC) of the test isolates against ciprofloxacin.

## PMQR prevalence

Remarkably, out of the 642 strains, as many as 414 (64.5%) harboured $aac(6')$-$Ib$-$cr$. The $qnrB$ and $qnrS$ genes were present in 97 (15%) and 64 (10%) isolates respectively. None of the strains were positive for $qnrA$ & $qnrD$, indicating the absence of these $qnr$ alleles among the clinical isolates included in the study (Fig. 2). The proportion of $aac(6')$-$Ib$-$cr$, $qnrB$ and $qnrS$ among the clinical isolates was found to be 64.49, 15.1 and 9.96 respectively. The overall prevalence of $aac(6')$-$Ib$-$cr$, $qnrB$ and $qnrS$ genes among the clinical Enterobacteriaceae strains isolated from JIPMER hospital could probably fall in the range of 60.72–68.12, 12.5–18.04 and 7.82–12.47 respectively.

E. coli had the maximum frequency of $aac(6')$-$Ib$-$cr$ (197) and $qnrB$ (43) genes. E. coli constituted almost half of the total $aac(6')$-$Ib$-$cr$ positive isolates. On the other hand, the frequency of the $qnrS$ gene was highest among K. pneumoniae isolates (32) with Klebsiella spp. altogether accounting for more than half of the total $qnrS$ gene identified (47). The most

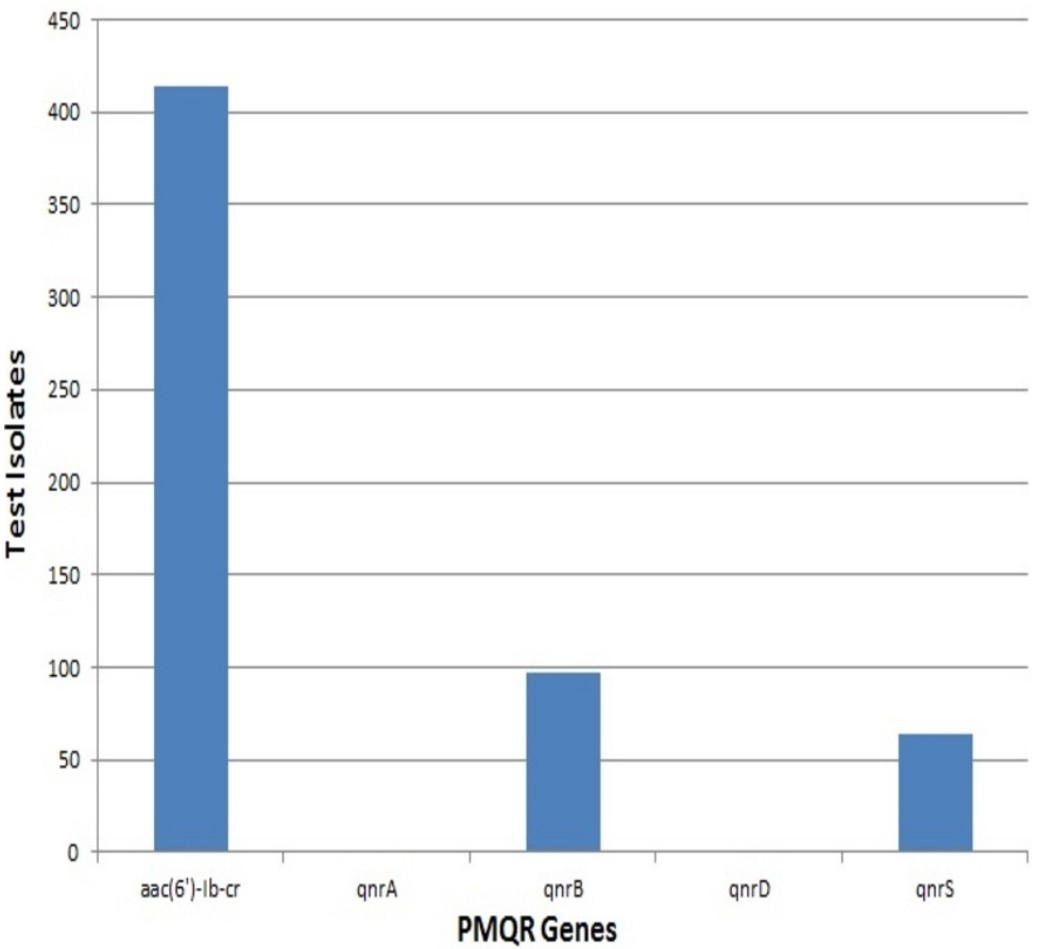

**Figure 2** Graphical representation of the prevalence of PMQR genes identified among the test isolates.

interesting fact was that PMQR genes were more predominant among *E. coli*, *Klebsiella spp.*, and *Enterobacter spp.* The occurrence of *qnr* alleles among *Citrobacter spp.*, *Serratia spp.*, *Proteus mirabilis* and *Providencia spp.* were almost negligible with *Proteus mirabilis* alone carrying the *qnrB* allele (Fig. 3). Remarkably, of the 106 test isolates with MIC >256 µg/mL only, three isolates were negative for PMQR genes.

The majority of the strains were found to carry one of the PMQR genes. But 7% of the isolates were found positive for multiple PMQR genes. All these isolates carried either *qnrB* or *qnrS* along with the *aac(6′)-Ib-cr* gene. None of the isolates harboured *qnrB* and *qnrS* simultaneously. *E. coli*, *Klebsiella spp.*, *Enterobacter spp.* and *Proteus mirabilis* were the organisms carrying multiple PMQR genes but, the association of *qnrS* with *aac(6′)-Ib-cr* was seen only in *E. coli* and *K. pneumoniae*. The presence of *aac(6′)-Ib-cr* and *qnrS* genes were associated with high MIC values ($\geq$64 µg/mL) and these associations were statistically significant with *p*-values <0.0000001 and <0.006261 respectively.

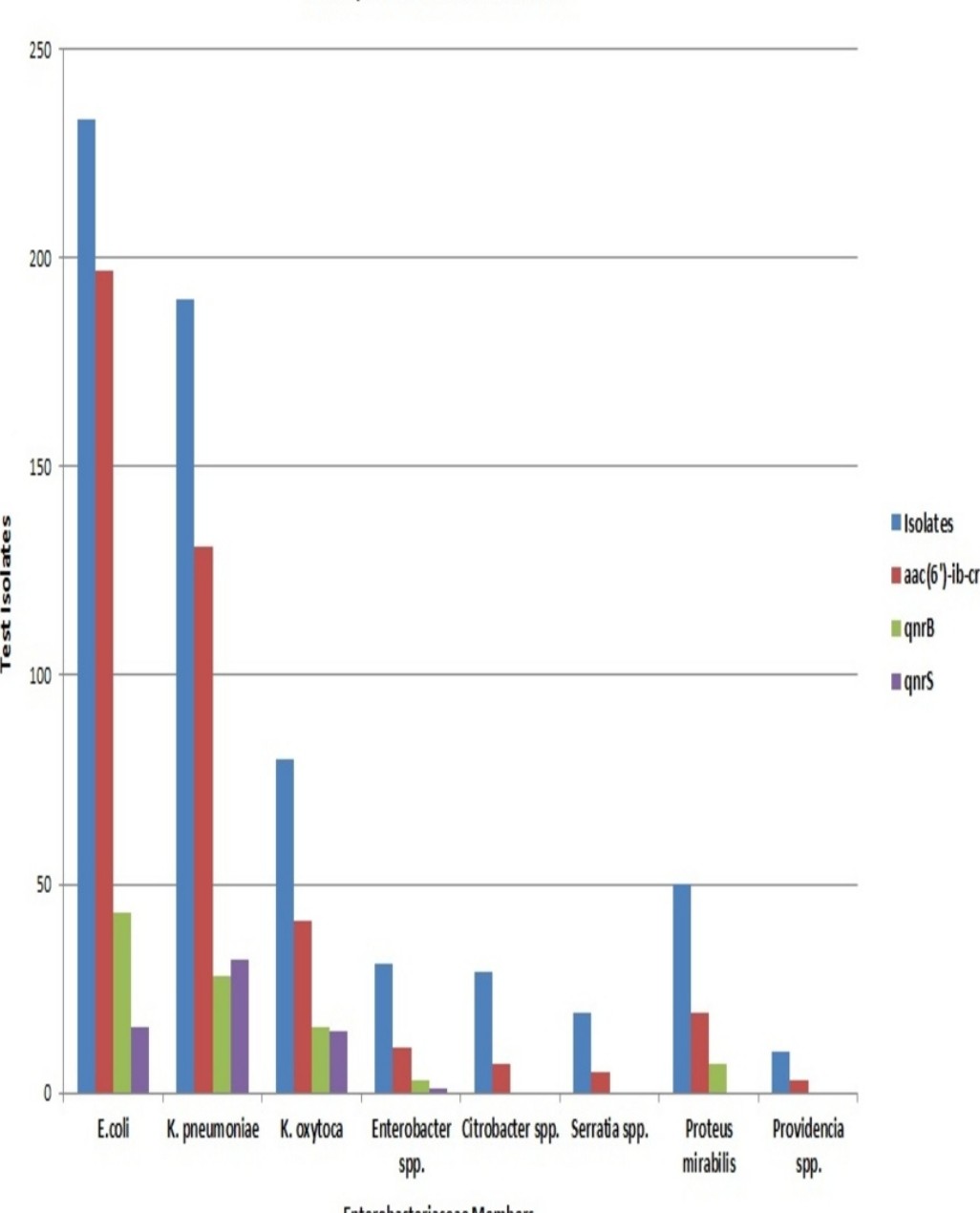

**Figure 3** Prevalence pattern of the identified PMQR genes across Enterobacteriaceae species among the test isolates.

Out of 528 PMQR positive isolates 212 were found to carry class I integron whereas, 95 isolates were found to carry class 2 integron and 47 isolates were positive for both the classes of integrons. However, we must admit that the study neither included the integron sequence analysis nor screened the integron-positive isolates for the presence of contiguous resistance gene cassettes.

## Transfer of PMQR

Of the 30 PMQR positive strains included in the conjugation only 18 transconjugants were successfully achieved. Among the transconjugants 11 were positive for *aac*(6′)-*Ib-cr*, four were positive for *qnrB*, and two were positive for *qnrS* genes. It is interesting to note that one particular transconjugant was found positive for both *aac*(6′)-*Ib-cr* and *qnrB*.

## Nucleotide sequencing

A total of 25 *aac*(6′) *Ib-cr*, seven *qnrB* and four *qnrS* strains were sequenced but none them had novel mutations within the nucleotide sequence. We have submitted nucleotide sequences of eight *aac*(6′) *Ib-cr*, two *qnrB* and one *qnrS* genes to GenBank and accession numbers assigned are: KR080535, KR080536, KR080537, KR080538, KR080539, KR080540, KR080541 and KR080543 for *aac*(6′)-*Ib-cr*, KR080544 & KR080545 for *qnrB* and KR080546 for *qnrS*. All the identified PMQR genes were found to be closely related based on the pair-wise distance matrix value. The overall distance matrix for *aac*(6′)-*Ib-cr* was found to be 2.642 whereas the pair-wise distance matrix for the *qnrB* gene sequences was found out be 1.255. The study did not attempt to identify variants of *qnr* genes. The pair-wise distance matrix of the *aac*(6′)-*Ib-cr* gene has been summarised as percentages, i.e., the number of nucleotide substitution per 100 nucleotide in the Data S2.

## DISCUSSION

The preceding decade has witnessed a very high usage of fluoroquinolones (*Geetha et al., 2014*). This extensive usage of fluoroquinolones has led to the emergence of Enterobacteriaceae isolates with reduced susceptibility to them. Interestingly, neither the fluoroquinolone prescription in the hospital nor the frequency of PMQR isolates varied much in the four years of the study (Fig. 4). In Enterobacteriaceae, the three major groups of *qnr* determinants are *qnrA*, *qnrB* and *qnrS* (*Geetha et al., 2014*) with *qnrD* having a prevalence of negligible extent. The *qnrC* gene was not included in this study as it has got the least prevalence (*Kim et al., 2009*).

The sample sizes of the previous studies from India concerning the prevalence of PMQR genes among Enterobacteriaceae ranged between 20 and 100 (*Pazhani et al., 2011*; *Geetha et al., 2014*; *Magesh et al., 2011*; *Bhattacharya et al., 2011*) . We have reported the prevalence of PMQR determinants from a sample size of 642 ciprofloxacin-resistant Enterobacteriaceae clinical isolate, making our report the largest study of its kind so far in India.

A high prevalence of resistance among Enterobacteriaceae against fluoroquinolones particularly ciprofloxacin has been demonstrated in this study similar to previous reports from all over the world. However, there are few striking differences. Firstly, a variety of Enterobacteriaceae species were included in this study (Fig. 3), compared to the previous reports that are mostly limited to *E. coli* and *Klebsiella pneumoniae* with rare inclusions of *Proteus spp.* and *Enterobacter spp.* (*Veldman et al., 2011*; *Wang et al., 2003*; *Pasom et al., 2013*; *Yang, Nam & Lee, 2014*). This study reports the presence of PMQR in *Providencia rettgeri* for the first time. Secondly, the proportion of *aac*(6′)-*Ib-cr* genes among our isolates was found to be high. The prevalence of *aac*(6′)-*Ib-cr* in previously published reports ranged between 7% and 40% (*Wang et al., 2003*; *Pasom et al., 2013*; *Veldman et al.,*

## Annual Fluoroquinolone Prescription & PMQR Isolation

**Figure 4** Graphical representation of the PMQR isolate annual frequency against ciprofoxacin tablets (500 mg) annual prescription in JIPMER hospital for consecutive four years.

*2011*); however, the prevalence rate of the same gene in our study was found to be 64.5%. The first PMQR gene to be identified and reported was *qnrA* (*Strahilevitz et al., 2009*) but, this allele was found to be absent among our clinical strains. It is remarkable to note that all the *E. coli* resistant to ciprofloxacin were found to carry one or the other PMQR gene. This is worrisome because PMQR genes are capable of horizontal transfer thereby accelerating the spread of this resistance mechanism among various clinical pathogens.

It is known that QRDR mutations induce high-level MICs while, PMQR genes induce low-level MICs (*Strahilevitz et al., 2009*; *Robicsek et al., 2006*). However, of the 528 PMQR-positive test isolates found in this study, 329 (62.2%) had MICs $\geq 64$ μg/mL, while only 19 (16.5%) of the 114 isolates lacking a PMQR gene had MICs $\geq 64$ μg/mL ($p < 0.0001$). Therefore, there was a significant association between increased MIC and the presence of PMQR genes, opening up the possibility that PMQR genes could have contributed to high MICs among the test isolates. But, we must agree that the efflux pump activities

of these test isolates were not elucidated and their QRDR mutation profile was also not identified. Thus, this particular finding of the study is inconclusive as to how far PMQR genes have contributed to the increase in MIC of a strain against ciprofloxacin. With future investigation of these clinical isolates for QRDR mutations and efflux mechanisms, the prominence of PMQR in fluoroquinolone resistance can be elucidated.

## CONCLUSION

In the present study, we have elucidated the prevalence of the plasmid-mediated quinolone resistance genes among clinical Enterobacteriaceae isolates recovered from a tertiary care hospital in Puducherry, India. Resistance to fluoroquinolones has predominantly increased with a majority of the isolates exhibiting high MIC values. Therefore, to combat this increased fluoroquinolone resistance it would be appropriate to use fluoroquinolones rationally for treating gram-negative infections. However, the significant finding of our study is that the prevalence of PMQR genes in JIPMER hospital is very high.

Moreover, the majority of the literatures on *qnr* gene are on prevalence rates from around the world and reports on mechanistic aspects at the molecular level are very few. Future research should focus more the molecular mechanism of the PMQR genes and its encoded proteins.

## ACKNOWLEDGEMENTS

The authors thank Dr. GA Jacoby, Lahey Clinic for providing us the *E. coli* strain J53 and Dr. T Ramamurthy, Translational Health Science and Technology Institute for providing *E. coli* (J53), *Shigella boydii* (IDH738), *E. coli* (BCH1108), *Morganella morganii* (500914) and *E. coli* (TC145) harbouring *qnrA*, *qnrB*, *qnrS*, *qnrD* & *aac(6')-Ib* and *qnrA* & *aac(6′)-Ib-cr* genes, respectively.

### Funding

This study was supported by JIPMER Institute Research Council intramural grant. The funders had no role in study design, data collection and analysis, decision to publish, or preparation of the manuscript.

### Grant Disclosures

The following grant information was disclosed by the authors:
JIPMER Institute Research Council intramural grant.

### Competing Interests

The authors declare there are no competing interest.

### Author Contributions

• Thiyagarajan Yugendran conceived and designed the experiments, performed the experiments, analyzed the data, contributed reagents/materials/analysis tools, wrote the paper, prepared figures and/or tables.

● Belgode Narasimha Harish conceived and designed the experiments, contributed reagents/materials/analysis tools, reviewed drafts of the paper.

## Ethics

The following information was supplied relating to ethical approvals (i.e., approving body and any reference numbers):

The study was approved by JIPMER Institute Ethics Committee, Jawaharlal Institute of Post Graduate Medical Education and Research, Puducherry, India (ECR/324/Inst/PY/2013).

## Data Availability

The raw data has been supplied as Supplemental Information.

## Supplemental Information

Supplemental information for this article can be found online at http://dx.doi.org/10.7717/peerj.1995#supplemental-information.

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
