# Peer review of "High incidence of plasmid-mediated quinolone resistance genes among ciprofloxacin-resistant clinical isolates of Enterobacteriaceae at a tertiary care hospital in Puducherry, India"

_PeerJ, doi:10.7717/peerj.1995_

## Round 0.1 · original submission · Major Revisions

· Academic Editor

Major Revisions

Please implement all comments from both reviewers.

Reviewer 1 ·

Basic reporting

Literature citation could be improved. The paper describing qnr identification is not cited- just a later review. The citations do not make clear to what extent PMQR prevalence studies have been carried out in India prior to this work.

Figure labelling is variable: The legend/caption for Figure 1 needs to make clear that these are ciprofloxacin MICs; lane labels and marker sizes are missing from Figure 2; Table S2 has no legend and needs to explain how the values shown were derived.

Experimental design

Mostly clear, but some details are missing, e.g. suppliers of reagents and discs for antibiotic susceptibility testing and citations for boiling lysis method and the MEGA software. In addition, Table S2 contains 25 sequences. However, only 10 accession numbers appear to be available for aac(6’)-Ib-cr genes. The paper does not make clear what fraction of identified genes were sequenced, and how many novel sequences were identified as a result.

Validity of the findings

This is a potentially useful study which extends knowledge of the prevalence of PMQR. The consideration of quinolone usage is also of interest. However, conclusions are not well presented- the Abstract (line 33) and Discussion (line 167) directly contradict one another with regard to prevalence trends; and the comment in the abstract about reconsideration of clinical breakpoints is not revisited at any point in the manuscript.

The significance of these findings is made harder to assess given that the comparison with other studies is very limited, and the construction of the manuscript does not make clear to which studies the authors refer; how much bigger the sample size is in this "large" study; or how much greater the prevalence of aac(6’)-Ib-cr really is. It would also be useful to provide some indication of how quinolone usage in the study facility compares with others elsewhere in India.

More could also be done to present the data in context. For example, most aspects of Figure 4 (frequencies with which individual species, resistance in strains of these, and individual genes in resistant strains are identified) are simply not described in the text. Information is also lacking about the nature of the facility in which the study was carried out.

Additional comments

In addition to the above, a number of other points should be clarified in a revised version of the manuscript.

Line 46: the original citation should be provided
Line 48: enzymic modification not degradation
Line 48: should introduce aac(6’)-Ib-cr here
Line 53: the meaning of His6 is not clear here. Remove.
Line 60: cite available studies?
Line 62: expand abbreviation (JIPMER) at first use
Line 62: useful to comment here on the nature of the JIPMER environment
Lines 74 – 76: useful to provide references for positive controls where these exist
Line 113: expand abbreviation (MDR) at first use
Line 125: delete “while”
Line 141: italics missing for species names
Line 159: Table S2 contains 25 sequences. However, only 10 accession numbers appear to be available for aac(6’)-Ib-cr genes. It is not clear what fraction of identified genes were sequenced.
Line 173: these reports should be cited
Line 174: what is a “large” sample size? The authors can be more specific about how the size of this study relates to those previously published.
Line 185: remove “vast number.” What is meant is that aac(6’)-Ib-cr was found in a high proportion of isolates studies.
Line 186: what were the frequencies identified in these previous reports? To which reports do the authors refer?
Lines 192 – 202: how do these values (proportion of isolates with elevated MICs carrying PMQR genes) compare to those from other similar studies?

·

Basic reporting

The manuscript is generally well presented and contains a discrete piece of work. It is an observational study so there is no hypothesis per se. Appropriate literature has been cited, focussing on local regional data, which is acceptable. Two of the figures can be deleted - as mentioned in general comments below.

Experimental design

There was a clear research question and the experimental design was appropriate to answer that question. This is a local study but it does not replicate something that has already been done, so far as I can tell.

Validity of the findings

I was uncertain of some statistical analyses as highlighted in the general comments below, but the findings appear valid and conclusions appropriate.

Additional comments

JIPMER should be defined in the main text at first use.

Line 64: this seems a very high percentage of resistance and the reference is "unpublished data". The authors should support this by reference to published work from other Indian studies.

Line 74: E. coli is the correct abbreviation. - where these isolates non-sequential; i.e. definitely not duplicates.

Line 83: MIC values also by CLSI methodology?

Line 88: by consumption you mean prescribing data?

Line 113: MDR is typically defined as resistant to 3 or more classes of antimicrobials. It is difficult given the limited number of antibiotics tested to say that more of the isolates were MDR. Perhaps should just say resistance to all antibiotics tested, rather than MDR.

Line 122: for UTI or for all indications?

Line 128: to what does the confidence interval here refer? You tested a group of isolates and determined that a certain percentage are positive for a gene. Where is the potental for error? At least you should explain the test performed to define the CI.

Line 142; is this a correlation coefficient or some sort of non-parametric statistics? Needs defining.

Line 150: which species were used to attempt transfer?


Line 164: it is also true to say that there is extensive contamination of the environment with quinolones in many parts of India - and elsewhere in the world - has anything been seen in that regard in the geographical region associated with this study?

Line 187: astonishing is a little strong here. It is likely that certain plasmid clone types can circulate in a given region, making certain resistance genes more or less common.

Fig 1: needs a Y axis and the legend bar is unnecessary.

Fig 2A and 2B are unnecessary and can be deleted.

---

## Round 0.2 · accepted · Accept

· Academic Editor

Accept

Thank you for following all review recommendations.